# Improvement of mRNA Delivery Efficiency to a T Cell Line by Modulating PEG-Lipid Content and Phospholipid Components of Lipid Nanoparticles

**DOI:** 10.3390/pharmaceutics13122097

**Published:** 2021-12-06

**Authors:** Hiroki Tanaka, Ryo Miyama, Yu Sakurai, Shinya Tamagawa, Yuta Nakai, Kota Tange, Hiroki Yoshioka, Hidetaka Akita

**Affiliations:** 1Laboratory of DDS Design and Drug Disposition, Graduate School of Pharmaceutical Sciences, Chiba University, 1-8-1 Inohana, Chuo-ku, Chiba City 260-0856, Japan; bfdref@gmail.com (R.M.); yu_sakurai@chiba-u.jp (Y.S.); 2DDS Research Laboratory, NOF CORPORATION, 3-3 Chidori-cho, Kawasaki-ku, Kawasaki City 210-0865, Japan; shinya_tamagawa@nof.co.jp (S.T.); yuta_nakai@nof.co.jp (Y.N.); kota_tange@nof.co.jp (K.T.); hiroki_yoshioka@nof.co.jp (H.Y.)

**Keywords:** mRNA delivery, lipid nanoparticles, T cells

## Abstract

(1) Background: T cells are important target cells, since they exert direct cytotoxic effects on infected/malignant cells, and affect the regulatory functions of other immune cells in a target antigen-specific manner. One of the current approaches for modifying the function of T cells is gene transfection by viral vectors. However, the insertion of the exogenous DNA molecules into the genome is attended by the risk of mutagenesis, especially when a transposon-based gene cassette is used. Based on this scenario, the transient expression of proteins by an in vitro-transcribed messenger RNA (IVT-mRNA) has become a subject of interest. The use of lipid nanoparticles (LNPs) for the transfection of IVT-mRNA is one of the more promising strategies for introducing exogenous genes. In this study, we report on the development of LNPs with transfection efficiencies that are comparable to that for electroporation in a T cell line (Jurkat cells). (2) Methods: Transfection efficiency was improved by optimizing the phospholipids and polyethylene glycol (PEG)-conjugated lipid components. (3) Results: Modification of the lipid composition resulted in the 221-fold increase in luciferase activity compared to a previously optimized formulation. Such a high transfection activity was due to the efficient uptake by clathrin/dynamin-dependent endocytosis and the relatively efficient escape into the cytoplasm at an early stage of endocytosis.

## 1. Introduction

Manipulation of the functions of immune cells by gene delivery is one of the more promising strategies in the field of medical science. Among the immune cells, T cells are important target cells since they can exert direct cytotoxic effects on infected/malignant cells, and play a role in the regulatory functions of other immune cells in a target antigen specific manner [1]. An ex vivo T cell therapy was recently developed based on the cell-killing activity of CD8+ T cells [2]. A typical example of such ex vivo T cell therapy is chimeric antigen receptor (CAR) introduced T cell (CAR-T) therapy [3]. The CAR is an artificial chimeric protein that is composed of a target recognition domain (single-chain variable fragment of an antibody) and intracellular T cell activating domains. The target of the CAR-T cells can be controlled by modifying the sequence of the target recognition domain. While early applications of such CAR-T cells were limited to B cell leukemia by targeting the CD19 protein on the tumor cells [4], applications are now being expanded to solid tumors or to other purposes such as the eradication of senescent cells from the body (senolysis) [5,6]. In addition to the CAR protein, other proteins have been widely investigated for further functional modifications of T cells. Representatives of such modifications include an enhancement in cytotoxic activity by avoiding T cell exhaustion [7], improved safety profiles by removing inherent T cell receptors from the T cell population [8], and the establishment of universal CAR-T cells by eliminating major histocompatibility complexes from their genome [8]. For these purposes, a highly efficient and safe technology for introducing exogenous proteins to the T cells is urgently needed.

Current approaches for the introduction of these exogenous genes to T cells have largely involved the use of viral vectors [9]. While these vectors can achieve efficient gene expression, the risk of the insertion of exogenous DNAs into the genome is always a possibility [10,11,12]. Moreover, the permanent alteration of the genome is attended by the additional risk of persistent adverse events [13]. As a solution to these problems, the transient expression of proteins by an in vitro-transcribed messenger RNA (IVT-mRNA) has attracted the interest of scientists in this field. Since IVT-mRNA is a highly vulnerable and hydrophilic molecule, its introduction into the T cells requires a delivery system.

One of the currently available methods for the transfection of the IVT-mRNA is electroporation [14]. Electroporation causes the perturbation of the plasma membrane using electric pulses. The IVT-mRNA can then be internalized into the cytoplasm through the pores generated by this perturbation. Electroporation allows the efficient and uniform introduction of the IVT-mRNA independent of endocytosis. On the other hand, electroporation could cause integrated stress responses and T cell exhaustion, probably triggered by the perturbation of the plasma membrane [15]. Transfection using lipid nanoparticles (LNPs) is an alternative strategy for introducing IVT-mRNA into cells. LNPs typically contain an artificial amphiphilic material referred as to a pH-sensitive cationic lipid or an ionizable lipid [16,17,18,19,20,21]. The head groups of ionizable lipids contain tertiary amine moieties and a backbone composed of hydrophobic scaffolds. Once taken up by the cells, the tertiary amine moiety can sense the lower pH of the endosomal compartment and then promote the cytoplasmic delivery of the genes and/or oligonucleotides. In 2018, Onpattro^®^, an siRNA formulated in an LNP, was approved as the first siRNA therapeutics [22]. More recently, RNA vaccine technologies against SARS-CoV-2 have been approved [23,24,25]. These historic successes of the therapeutic applications of the LNPs promise to accelerate further applications of the LNPs in the biological and medical field.

Regarding the delivery of nucleic acids, it is generally considered that T cells are negligibly transfected in comparison with other possible targets; both immune cells (macrophages or dendritic cells) and non-immune cells (hepatocytes or tumor cells) [26,27]. One plausible reason for this difference is that the activity of endocytosis of the T cells is poor [28,29]. To achieve successful transfection to the T cells, the development of an LNP that can be taken up by the T cells is prerequisite. The interaction between the LNPs and the cells is affected by the surface properties of the LNPs. Therefore, lipid components, especially those with hydrophilic moieties, would be expected to significantly affect transfection efficiency. In this study, we report on the development of LNPs that show a transfection efficiency comparable to that for electroporation in a T cell line (Jurkat cells) without cellular stress responses and toxicity. In this process, the type of phospholipids, and their composition, in parallel with polyethylene glycol (PEG)-conjugated lipids was optimized.

## 2. Materials and Methods

### 2.1. Materials

Detailed supplier’s information including item numbers of all reagents used in this study are listed in Appendix A (Appendix A). As a component of LNPs, a self-degradable lipid-like material was used (ssPalmO-Phe-P4C2) [30]. The ssPalmO-Phe-P4C2 (Product# COATSOME^®^ SS-OP), 1,2-dioleoyl-sn-glycero-3-phosphatidylcholine (DOPC, Product# COATSOME^®^ MC-8181), 1-palmitoyl-2-oleoyl-sn-glycero-3-phosphatidylcholine (POPC, Product # COATSOME^®^ MC-6081), 1-palmitoyl-2-oleoyl-sn-glycero-3-phosphatidylethanolamine (POPE, Product # COATSOME^®^ ME-6081), and 1-(Monomethoxy polyethyleneglycol2000)2,3-dimyristoylglycerol (DMG-PEG2000, Product # SUNBRIGHT^®^ GM-020) were supplied by NOF CORPORATION (Tokyo, Japan). The information of the stock solution of lipids is summarized in the supporting materials (Appendix A). The IVT-mRNAs coding reporter genes (firefly luciferase and EGFP) were purchased from TriLink BioTechnologies (San Diego, CA, USA). Quant-IT™ RiboGreen^®^ RNA reagent and Lipofectamine^®^ MessengerMAX were purchased from ThermoFisher Scientific (Waltham, MA, USA). Pitstop^®^ 2 Novel cell-permeable clathrin inhibitor and Dynole^®^ 34-2 dynamin I and dynamin II inhibitor were purchased from Abcam plc (Cambridge, UK). InSolution™ Cytochalasin D and cholesterol was purchased from Sigma-Aldrich (St. Louis, MO, USA). Genistein was purchased from Santa Cruz Biotechnology (Dallas, TX, USA). All other reagents and chemicals were commercially available and were used without further purification.

### 2.2. Cell Culture

Jurkat cells were cultured using RPMI 1640 medium supplemented with 10% (*v/v*) FBS, 1 mM sodium pyruvate, 10 mM HEPES, 100 U/mL of penicillin, 4.5 g/L glucose and 100 mg/mL of streptomycin. Cells were cultured in 10 cm dishes (Asnol Sterilization Petri Dish φ90 × 15 mm, AS ONE Corporation, Osaka, Japan) and were passed when they reached 80% confluence. Typical passage timing was at 3-day intervals. The cells were cultured under an atmosphere of 5% CO_2_/air at 37 °C.

### 2.3. Preparation of the mRNA-LNP

The IVT-mRNA encoding luciferase or EGFP was diluted with a 20 mM malic acid buffer (30 mM NaCl, pH 3.0) at a concentration of 0.0067 µg/µL. The lipid ethanol solution was prepared at a concentration of 4 mM. The detailed description for the preparation of the lipid mixtures is summarized in the supporting materials (Appendix A). These solutions were mixed using a microfluidics device (NanoAssemblr^TM^, Precision NanoSystems, Vancouver, BC, Canada) (total flow rate; 4 mL/min, flow ratio; water/ethanol = 3/1 (*v/v*)). The mixture of the mRNA and lipids (0.8 mL) were recovered and diluted with 3 mL of MES/NaOH buffer (2-(N-morpholino) ethanesulfonic acid, 20 mM, pH6.5). The external solution was replaced with PBS(-) by ultrafiltration using Amicon Ultra-4-100K centrifugal units. The particle solution was diluted to an adequate concentration with PBS(-) before transfection.

### 2.4. Characterization of the Particles

Size, Polydispersity Index (PdI), and Zeta-potential of LNPs were measured by dynamic light scattering (Zetasizer nano ZS, Malvern Panalytical, Malvern, UK). Encapsulation efficiency of the IVT-mRNA were obtained by Quant-IT^TM^ RiboGreen^®^ assay [30].

### 2.5. mRNA Transfection and In Vitro Luciferase Assay

#### 2.5.1. Transfection by Electroporation (EP)

Jurkat cells were resuspended in Opti-MEM (1.11 × 10^7^ cells/µL, 90 µL). An mRNA solution was diluted in Opti-MEM (0.2 µg/µL, 10 µL). The mRNA solution was then added to the cell suspension and set to an electroporator NEPA21 Type II (Nepa Gene Co., Ltd., Chiba, Japan). The electroporation was conducted at a Pp (poring pulse) of 175 V; a Pp duration of 2.5 ms; a Pp interval of 50 ms; a Pp frequency of 2 times; a Pp decay rate of 10%; a Tp (Transfer pulse) of ±20 V; a Tp duration of 50 ms; a Tp interval of 50 ms; a Tp frequency of 5 times for ± Voltage; a Tp decay rate of 40%. Immediately after the electric pulses, the cells were collected by adding 300 µL of culture media. The cells were incubated for 1 h at 37 °C in the CO_2_ incubator. After the recovery culture, the cells were collected by centrifugation (4 °C, 500× *g*, 3 min), and then resuspended at a concentration of 1 × 10^5^ cells/mL in culture media containing serum. The cells were transferred to 3.5 cm dishes (1 × 10^5^ cells/mL, 2 mL).

#### 2.5.2. Transfection by Lipofectamine MessengerMAX (LFN)

The transfection was conducted according to the manufacture’s protocol. Jurkat cells (1 × 10^5^ cells/mL, 2 mL) were seeded in 3.5 cm dishes. Lipofectamine MessengerMAX reagent (ThermoFisher Scientific, Waltham, MA, USA) and the mRNA were mixed in the Opti-MEM (ThermoFisher Scientific, Waltham, MA, USA) at a ratio of 1.5 µL reagent/µg mRNA (0.5 µg). The mixture was incubated for 5 min. After the incubation, the mixture was transfected to the Jurkat cells in culture media containing serum.

#### 2.5.3. Transfection by Lipid Nanoparticles (LNPs)

Jurkat cells (1 × 10^5^ cells/mL, 2 mL) were seeded in 3.5 cm dishes. The LNPs containing 0.4 µg of mRNA was added to the Jurkat cells in culture media containing serum.

#### 2.5.4. Luciferase Assay with Incubator-Type Luminometer

In all samples (EP, LFN, and LNP), final concentration of the mRNA and the cells were 0.2 µg/mL and 1 × 10^5^ cells/mL. The cells were incubated in culture media containing 100 µM of D-luciferin potassium on the 3.5 cm dishes. The cells were placed in an incubator-type luminometer (Kronos, ATTO, Tokyo, Japan) and the luciferase activity was measured for 2 min at 1 h intervals.

### 2.6. Analysis of Phosphorylation of eIF2α

The IVT-mRNA was transfected to the Jurkat cells by EP or LNPs as described in Section 2.5. At 1, 2, 4 or 6 h after transfection, the cells were collected and washed with PBS(-). The cells were fixed by treatment with 100 µL of 4% paraformaldehyde and then washed with PBS(-). The cell membranes were permeabilized by treatment with 100 µL of Methanol (4 °C). The cells were washed twice with FACS buffer (0.5% BSA, 0.1% NaN_3_ in PBS(-)), and phosphorylated eIF2α was reacted with a primary rabbit-a-eIF2a-P antibody (Rabbit monoclonal [E90] to EIF2S1 (phospho S51), Abcam, ab32157, 100-fold dilution) for 45 min. The cells were washed twice with the FACS buffer, and then stained with secondary antibody (Donkey anti-Rabbit IgG (H+L) Highly Cross-Adsorbed Secondary Antibody, Alexa Fluor 647, Invitrogen^TM^, A-31573, 100-fold dilution, ThermoFisher Scientific, Waltham, MA, USA). The cells were washed twice, and the cells were re-fixed with 1% paraformaldehyde. The fluorescence of the Alexa Fluor 647 was measured by Novocyte^TM^ flow cytometer (Agilent, Santa Clara, CA, USA).

### 2.7. Analysis of Cellular Uptake

#### 2.7.1. Analysis of the LNPs with Different Composition

LNPs without mRNA were prepared with different amounts of PEG-lipids (0.375%, 0.75%, 1.5%, or 3%) or phospholipids (DOPC, POPC, or POPE). The LNPs were labelled with 0.2% of DiD fluorescent dye. The cells were pre-incubated for 30 min at 37 °C or on ice. Jurkat cells were transfected with the LNPs at a concentration of 40 nmol/mL total lipids (total of ssPalmO-Phe-P4C2, phospholipids, and cholesterol). The cells were incubated for 3 h at 37 °C or on ice. After incubation, the cells were collected and washed with 500 µL of FACS buffer twice. Fluorescence of the DiD from the cells was measured by the Novocyte^TM^ flow cytometer (Agilent, Santa Clara, CA, USA).

#### 2.7.2. Cellular Uptake of the LNPs in the Presence of an Endocytosis Inhibitor

For the pre-treatment of the cells, the cells were seeded at 1 × 10^5^ cells/mL in serum-free RPMI-1640 medium. The cells were incubated with 30 µM Pitstop, 20 µM Dynole, 100 µM Genistein, or 10 µM Cytochalasin D for 20 min. After the incubation, the cell suspension was centrifuged (4 °C, 500× *g*, 5 min) and 1 mL of culture medium containing the same concentration of the inhibitors was added to the pelleted cells. Cells were transfected with the DiD-labelled LNPs without mRNA at a concentration of 40 nmol/mL total lipids (equivalent to the transfection of 0.2 µg mRNA) for 1 h. After transfection, the cells were collected and washed with 500 µL of the FACS buffer. Fluorescence of the DiD from the cells was measured by means of a Novocyte^TM^ flow cytometer (Agilent, Santa Clara, CA, USA).

### 2.8. Hemolysis Assay

Whole blood of ICR mice was collected from inferior vena cava in the presence of 0.5 μL of heparin sodium (1000 U/mL). The ICR mice (male, 6–7 weeks) were purchased from Japan SLC, Inc. (Shizuoka, Japan). The experimental protocols were reviewed and approved by the Chiba University Animal Care Committee in accordance with the “Guide for Care and Use of Laboratory Animals”. The ethical approval code issued from the committee was 30-41. Red blood cells were purified by washing the blood (1 mL) with 9 mL of PBS. The blood was centrifuged (4 °C, 400× *g*, 5 min) and the supernatant was discarded by aspiration. The washing was repeated 5 times so as to completely remove serum proteins. PMBS buffer was prepared by dissolving DL-malic acid by PBS(-). Final concentration of the malic acid was 20 mM. The pH was then adjusted by NaOH solution. The red blood cells were then incubated with the LNP at pH from 5.5 to 7.4. The final concentration of the total lipid was 100 µM. The mixture was incubated at 37 °C for 30 min. After the incubation, the samples were centrifuged at 500× *g* for 5 min. The absorbance of the leaked hemoglobin in supernatant was measured at 545 nm by plate reader Infinite 200 (Tecan Japan Co., Ltd., Kanagawa, Japan). As a positive control, the red blood cells were lysed by 0.1% (*wt*/*v*) Triton-X100.

## 3. Results

### 3.1. Surface of the LNP Affected to the Transfection Efficiency

As a main component of LNPs, an SS-cleavable and pH-activated lipid-like material (ssPalmO-Phe-P4C2) was used (Figure 1). Tertiary amine moieties and a disulfide bond in the ssPalmO-Phe-P4C2 promote endosomal escape and the cytoplasmic release of the mRNA, respectively [31]. The use of a combination of a disulfide bond and phenyl esters further improved the cytoplasmic release of the IVT-mRNA via an intraparticle hydrolytic reaction [30]. A commercially available luciferase IVT-mRNA (described in 2.1. materials) was used as a reporter gene. The LNP_ssPalm_ formulated with the ssPalmO-Phe-P4C2 contained phospholipids, cholesterol, and polyethylene glycol (PEG)-conjugated lipids as helper lipids (Figure 1). It was previously reported that the LNP_ssPalm_ with a lipid composition of ssPalmO-Phe-P4C2/dioleoyl-sn-glycero-phosphatidyl choline (DOPC)/cholesterol/DMG-PEG2000 = 52.5/7.5/40/3 could be used to introduce the IVT-mRNA into HeLa cells (human cervical cancer), CT26 cells (mouse colon carcinoma), Hepa-1c1c7 cells (mouse liver carcinoma), and MEF cells (mouse embryonic fibroblast) [30]. However, this composition showed negligible transfection activity against the Jurkat cells. The ssPalm was a highly hydrophobic molecule and could be dispersed in a water phase only with the aid of the other lipids at a neutral pH [32]. It can, therefore, be concluded that the LNP_ssPalm_ was prevented from aggregation by the surface stabilized with phospholipids and DMG-PEG2000. Since this surface appears to be a key interface that regulates interactions with cells, the properties of this surface were modified by changing the content of phospholipids and the amount of PEG-lipid used in their preparation.

After a preliminary screening, we found that the use of 1-palmitoyl-2-oleoyl-phosphatidyl ethanolamine (POPE) as a lipid component improved the transfection activity regarding Jurkat cells. An optimized lipid composition of the LNP_ssPalm_ was ssPalmO-Phe-P4C2/POPE/Chol = 50/15/35. To evaluate the effects of the surface density of PEG, the amount of the PEG-lipid was changed from 3% to 0.375% of the total of other lipids. The properties of the LNP_ssPalm_ particles containing luciferase-mRNA are summarized in Table 1. The increase in the PEG resulted in the formulation of a smaller particle, while the changes in Polydispersity index (PdI), Zeta potential, and encapsulation efficiency were marginal. The transfection activity of the LNP_ssPalm_ with various densities of DMG-PEG2000 and various types of phospholipids was then compared (Figure 2a). As a result, LNPs with POPE showed a higher transfection activity than the LNPs with DOPC regardless of the PEG-lipid content. The most critical finding was that the incorporation of POPE and DMG-PEG2000 at a concentration below 0.75% showed a high transfection efficiency for the Jurkat cells. Since the chemical structure of POPE and DOPC were different in both hydrophobic scaffolds and hydrophilic heads, it was not possible to identify the important structural unit from this comparison. To clarify the effect of head groups of the phospholipids, the LNPs containing POPE were compared to LNPs containing 1-palmitoyl-2-oleyol-phosphatidyl choline (POPC). The comparison indicated that the phosphatidyl ethanolamine group in the POPE is important for the improved transfection efficiency (Figure 2b). In the following sections the composition of the LNP was fixed to ssPalmO-Phe-P4C2/POPE/Chol/PEG-lipids = 50/15/35/0.75, since this combination represented the maximum transfection activity.

### 3.2. The LNP_ssPalm_ Enables Comparable Transfection Activity to the Electroporation without Stress

The transfection efficiency of the LNP_ssPalm_ containing POPE was compared to that for electroporation (EP) and the lipofectamine messengerMAX (LFN). The expression of the luciferase mRNA was evaluated by an incubator type luminometer (Figure 3a). The EP showed higher luciferase expression at early times of the transfection. However, the attenuation in the luciferase expression of the EP was faster than that for the LNPs. A comparison of the cumulative luciferase activity revealed that the transfection efficiency of the LNPs was significantly higher than the LFN and was comparable to that for the EP (Figure 3b). The LNP showed no toxicity against Jurkat cells regardless of the IVT-mRNA encapsulation (Appendix A). Both EP and LNP_ssPalm_ introduced homogenous IVT-mRNA into the Jurkat cells as evaluated by the transfection of EGFP (Appendix A).

It was previously reported that electroporation induced integrated-stress-responses (ISRs) [15], an adaptive reaction against diverse cellular stresses such as unfolded protein responses, deprivation of nutrients, mitochondrial damages, and viral infections [33]. The induction of the ISRs results in the phosphorylation of an α-subunit of eukaryotic translation initiation factor 2 (eIF2α). Since the phosphorylation of eIF2α results in the shutdown of cap-dependent translation, ISRs should be avoided during the delivery of the IVT-mRNA. To evaluate the extent of ISRs, the phosphorylation level of the eIF2α was compared by flow-cytometry (Figure 4) [34]. Compared to the EP, which caused a significant increase in the phosphorylation of the eIF2α, no elevation was observed for the transfection with the LNP_ssPalm_. These observations indicate that the LNP_ssPalm_ with a composition of ssPalmO-Phe-P4C2/POPE/Chol/DMG-PEG02000 = 50/15/35/0.75 is a promising mRNA delivery system and represents an alternative to EP.

### 3.3. The Membrane Destabilizing Activity Was Enhanced by the Surface Modification

For the delivery of nucleic acids into the cytoplasm, the most important biological barrier is the plasma/endosomal membrane [35]. Endosomal escape efficiency could be evaluated by a hemolysis assay. Although the original purpose of the hemolysis assay was to evaluate the toxicity of hemolytic small drugs, hemolysis activity is frequently taken as an index of membrane destabilizing activity in the field of nucleic acid delivery since the anionic membrane of the red blood cells is a good model for the endosomal membrane [36,37]. The hemolysis activity of the LNP_ssPalm_ containing different amounts of the DMG-PEG2000 was first compared at pH 7.4 (physiological condition), pH 6.5 (early endosome condition), and pH 5.5 (late endosome condition) (Appendix A). The amount of DMG-PEG2000 ranged from 0.75% to 3%. At pH 5.5, all of the samples showed a high hemolysis activity and no significant differences were detected among the samples. At pH 6.5 and pH 7.4, the LNP_ssPalm_ with a small amount of the DMG-PEG2000 tended to show a higher hemolytic activity but the overall efficiency was poor. Therefore, the difference between 1.5% DMG-PEG2000 and 0.75% DMG-PEG2000 was evaluated in a narrower range of pH, i.e., from pH 5.5 to pH 6.5 (Figure 4). Among the tested pH values, the LNP_ssPalm_ with the 0.75% DMG-PEG2000 showed a slightly, but significantly higher hemolytic activity at pH 6.3 and pH 6.1 (Figure 5). Since the hemolysis activity was affected by an apparent pKa of the surface of LNPs, the pKa of the LNP_ssPalm_ with 0.75%, 1.5%, or 3.0% of the DMG-PEG2000 was evaluated by means of a 6-(p-Toluidino)-2-naphthalenesulfonic acid (TNS) assay. The TNS is a fluorescent dye that shows strong fluorescence when the dye is inserted to the hydrophobic environment. Since the TNS has an anionic charge, the dye can interact with the surface of an LNP depending on the cationic charge that develops in the acidic environment. The TNS assay showed that the PEG-lipid content had no effect on the apparent pKa of the LNP_ssPalm_ (Appendix A). Thus, the difference in the hemolysis activity collectively indicated that the decrease in the surface density of PEG-lipid facilitated the physical interaction between LNPs and the biological membrane. To clarify the effect of head groups, the difference between POPC and POPE was evaluated. The comparison also revealed that the LNP_ssPalm_ with POPE showed a slightly but significantly higher hemolytic activity at pH 6.3 and pH 6.1 (Appendix A). These data indicate that the LNP_ssPalm_ with the composition of ssPalmO-Phe-P4C2/POPE/cholesterol/DMG-PEG = 50/15/35/0.75 enhanced the endosomal escape at an early stage of endocytosis.

### 3.4. The LNP_ssPalm_ Was Taken Up by the Jurkat Cells via Clathrin/Dynamin-Dependent Endocytosis

Since the T cells generally have poor cellular uptake activity, the mechanism responsible for the efficient transfection activity by LNP_ssPalm_ is an important information. Since the uptake of the LNP_ssPalm_ was almost completely inhibited when the incubation was conducted at 4 °C, the LNP_ssPalm_ was assumed to be taken up by the Jurkat cells in an energy-dependent manner. (Figure 6a). The detailed uptake mechanism was evaluated by using pharmacological inhibitors of endocytosis. In this study, Pitstop^®^ 2, Dynole^®^ 34-2, Genistein, and Cytochalasin D were used as inhibitors for clathrin-dependent pathway, the dynamin-dependent pathway, the tyrosine kinase-dependent pathway, and the actin-dependent pathway, respectively. In the case of genistein, no decrease in cellular uptake was found. On the other hand, Pitstop and Dynole inhibited cellular uptake by 89% and 76%, respectively (Figure 6b). These data clearly show that the LNP_ssPalm_ was taken up by the Jurkat cells via clathrin/dynamin-dependent endocytosis. While a significant difference was observed in the case of the cytochalasin D treatment, the contribution of the actin molecule was up to 12% and was much less than that of clathrin and dynamin.

## 4. Discussion

It was reported that the LNPs made with an ionizable lipid, the lipids are distributed in an asymmetric manner; the hydrophobic core of the ionizable lipid was surrounded by the hydrophilic shell made with amphiphilic molecules such as phospholipids and PEG-lipids [38]. Since the LNP_ssPalm_ showed similar structural features such as an electron-dense spherical structure [30], we hypothesized that the surface of the LNP_ssPalm_ was also stabilized by these amphiphilic components in a similar manner. The phospholipid content and the density of the PEG-lipid were modified to investigate the effects of the surface property on the transfection efficiency to Jurkat cells. These modifications resulted in a 221-fold increase in luciferase activity from the LNPs containing DOPC with 3% DMG-PEG2000 compared to LNPs containing POPE with 0.75% DMG-PEG2000 (Figure 2a). This transfection activity was comparable to that obtained by electroporation. The reduction in the PEG-lipid from 3% to 0.75% contributed to a 42.7-fold increase in transfection activity. A comparison between “DOPC and POPE” (8.14-fold increase, Figure 2a) or “POPC and POPE” (5.86-fold increase, Figure 2b) also indicated that the employment of a helper lipid with a phosphoethanolamine structure in the head group is important in terms of maximizing transfection activity. We conclude that the effect of the PEG-lipid was more significant than that of the phospholipids. In the 8.27-fold increase in transfection efficiency that was achieved by changing the PEG-lipid from 1.5% to 0.75% (Figure 2a), the cellular uptake of the LNPs accounted for only a 1.8-fold increase (Figure 5a). Thus, the 5.30-fold increase in transfection efficiency can be attributed to intracellular trafficking after endocytosis. Since the PEG on the surface of the LNP_ssPalm_ had no effect on responsiveness to pH (Appendix A), the decrease in the PEG-lipid probably facilitated a physical interaction between the LNP_ssPalm_ and the biological membrane, and subsequently enhanced the cytoplasmic delivery of the IVT-mRNA. The decrease in the PEG-lipid also improved transfection activity against Raw264.7 cells (mouse macrophages/monocytes) and HeLa cells (human cervical cancer cells) (Appendix A). It is noteworthy that the effects of the helper lipid on the transfection activity were marginal for these cell lines. Although the overall contribution of changing the helper lipids was smaller than that for the PEG-lipid, the requirement of the phosphatidyl ethanolamine for the transfection to Jurkat cells indicated that the required properties for the nanoparticles for successful transfection to the T cells are different from these cells with high phagocytosis/micropinocytosis capacities. This observation also indicated that the optimization of the helper lipid content should be performed for the specific target cells being used. During the endocytosis process, receptors on T cells such as TCR, CD4, and CTLA-4 are recycled to the cellular surface via an endocytic recycling compartment. The recycling pathway is also employed to replenish the newly synthesized perforin in cytotoxic T cells. Thus, the recycling pathway plays a key role in maintaining the function of T cells [39]. The pH of recycling endosomes decreases to approximately pH 6.5. Thus, even when LNPs are sorted into the recycling compartment, the escape from this compartment requires membrane destabilizing activity under weakly acidic conditions. Actually, in addition to the poor cellular uptake, the slow rate of the acidification of endosomal compartment was reported to be another biological barrier for the delivery of genes to the T cells by pH sensitive materials [40]. The enhancement in transfection activity indicated that the incorporation of POPE and the use of 0.75% DMG-PEG2000 could overcome these barriers. The enhancement in transection efficiency reported here could be explained, at least in part, by the enhanced membrane destabilizing activity of the composition, especially at pH 6.1 and pH 6.3 (Figure 5 and Appendix A), as well as the increased cellular uptake by the decreased level of surface PEG-lipids (Figure 6a). The importance of the endosomal escape at an early stage of endocytosis for the LNP-based mRNA delivery was recently reported as a preprint [41]. In this preprint, it was suggested that endosomal escape which is sufficiently productive to induce protein expression from loaded mRNA occurred mainly from the early endosomal compartment and the recycling compartment, as evidenced by super-resolution microscopy. This observation strongly supports our observations. Since the analysis of intracellular trafficking in the Jurkat cells was difficult because of its low level of cytoplasm and floating nature, evidence of endosomal escape at an early stage of the endocytosis was not clearly shown in this study. However, the hemolysis activity in the pH 6.1–pH 6.3 region could be used as an index for the further development of LNPs. The limitation of the hemolysis assay was that the contribution of proteins cannot be evaluated. A hemolysis assay in the presence of serum would be a more accurate index for the expected efficiency of LNPs [42].

Based on experiments involving pharmacological inhibitors of endocytosis, the findings revealed that the LNP_ssPalm_ particles were internalized into the Jurkat cells via clathrin-mediated endocytosis (CME). This CME is a common uptake pathway and plays a key role in nutrient uptake, receptor signaling, and vesicle recycling [43]. In the case of T cells, clathrin-mediated endocytosis would be expected to be the most active pathway [6]. The size of the clathrin-coated pit was reported to be less than 200 nm [44]. Since the size of the LNP_ssPalm_ was below 100 nm, the contribution of this clathrin-mediated pathway is reasonable. Since the transfection activity of the LNP_ssPalm_ was also inhibited by the inhibitor of CME, the contribution of this pathway to the successful cytoplasmic delivery was confirmed (Appendix A). It was reported that CME is important for the delivery of siRNA [45]. In this report, the down-regulation of the clathrin heavy chain reduced the uptake of the LNP by about half. It was reported that the inhibition of the budding of clathrin-coated particle by the cytosolic acidification and by treatment with chlorpromazine treatment strongly inhibits the cellular uptake of the LNP [46]. It was also suggested that the infection by viral particles requires CME [47]. Taking these reports into account, the internalization of the LNP_ssPalm_ via CME might be an important feature for the successful delivery of the cargo into the cell. In general, CME is known to be a receptor-dependent specific mechanism. We thus hypothesize that the LNP_ssPalm_ interacted with the surface protein(s) of the T cells, while the specific target and manner of the interaction remains to be clarified. Our experiments indicate that phagocytosis or macropinocytosis, the targets of genistein and cytochalasin D, respectively, are not important as the CME for T cells. This character of the T cells is one of the reasons for the notion that LNPs are poorly taken up by T cells.

In this study, the level of the phosphorylation of eIF2α was investigated as an index of ISRs. A schematic illustration and the importance of the ISRs are summarized in Appendix A. The induction of the ISRs in T cells are important in cancer immunotherapy. The expression of a stress related-transcription factor C/EBP homologous protein (CHOP), a consequence of protein kinase RNA-like endoplasmic reticulum kinase (PERK)-dependent and tumor-induced ISRs, results in a decreased cytotoxicity of T cells [48]. Thus, the ISRs in T cells has adverse effects on, not only the extent of protein expression, but also on the efficacy of T cell therapy. In the case of electroporation, the ISRs were induced by general control nonderepressible 2 (GCN2) and PERK [15]. GCN2 is a kinase which senses an increase in the levels of amino acid-uncharged t-RNA. The uncharged t-RNA increases in response to the decrease in available nutrients under the conditions such as serum depletion [49,50]. PERK is a kinase that senses the unfolded protein in the endoplasmic reticulum [51]. The activation of GCN2 and PERK by electroporation may indicate that the pore formation on these biological membranes results in the leakage of the contents of cytoplasm/endoplasmic reticulum into the extracellular space. This leakage would be expected to result in the disruption in the homeostasis of cells. The non-phosphorylation of eIF2α in the LNP_ssPalm_ incubation indicated that the disturbance of the homeostasis by the LNP_ssPalm_ was not significant, since the damage of the biological membrane occurs only in the acidic endosomal compartment in the case of the LNP_ssPalm_. Since the LNP_ssPalm_ showed no ISRs against Jurkat cells, we conclude that the LNP_ssPalm_ represents a viable alternative to electroporation. However, it is known that a high level of leakage of endosomal contents causes the apoptosis or necrosis [52,53,54]. Careful investigations of the effects of the LNPs on such cells is still needed.

Recently, the successful induction of the human CAR-T cells by LNP-based mRNA delivery was reported [55]. It was shown that the viability of T cells treated with these LNPs was superior to that treated with electroporation. These observations confirm that LNPs represent a safe alternative to the conventional electroporation method. In that report, LNPs that showed a high transfection activity for Jurkat cells could also introduce mRNA into primary human T cells. This indicates that Jurkat cells can work as a surrogate for the evaluation of transfection efficiency of a LNP system for T cells. On the other hand, it has been also known that properties such as signal transduction of immortalized T cells are different from those of activated primary T cells [56]. In addition to this point, our study investigated the efficiency of delivery of mRNA, but the effects of LNPs on the original cell-killing function of T cells remain to be clarified. For these reasons, the usefulness of the LNP system should be evaluated using primary T cells to make a clear conclusion. Especially, the importance of the uptake via CME and avoidance of ISRs observed in this study should be evaluated by functional evaluation using a primary culture system.

## 5. Conclusions

In this study, we report on the development of an LNP_ssPalm_ with the composition of ssPalmO-Phe-P4C2/POPE/cholesterol/DMG-PEG = 50/15/35/0.75 for the delivery of IVT-mRNA to Jurkat cells. The particles were taken up by clathrin/dynamin-dependent endocytosis. This composition showed membrane destabilizing activity at a slightly acidic pH from pH 6.1-6.3. It was thus hypothesized that hemolytic activity in this pH range can be used as an index for evaluating the cytoplasmic delivery of mRNA to the T cells. Since the transfection with the LNP_ssPalm_ showed no stress responses in the Jurkat cells, it would appear that the LNP_ssPalm_ represents a safe alternative technology for introducing proteins into such cells.

## 6. Patents

H. Tanaka, Y. Nakai, K. Tange, H. Yoshioka, and H. Akita are the inventors of the patent pending (WO2019/188867) on the ssPalm chemicals.

## Figures and Tables

**Figure 1 pharmaceutics-13-02097-f001:**
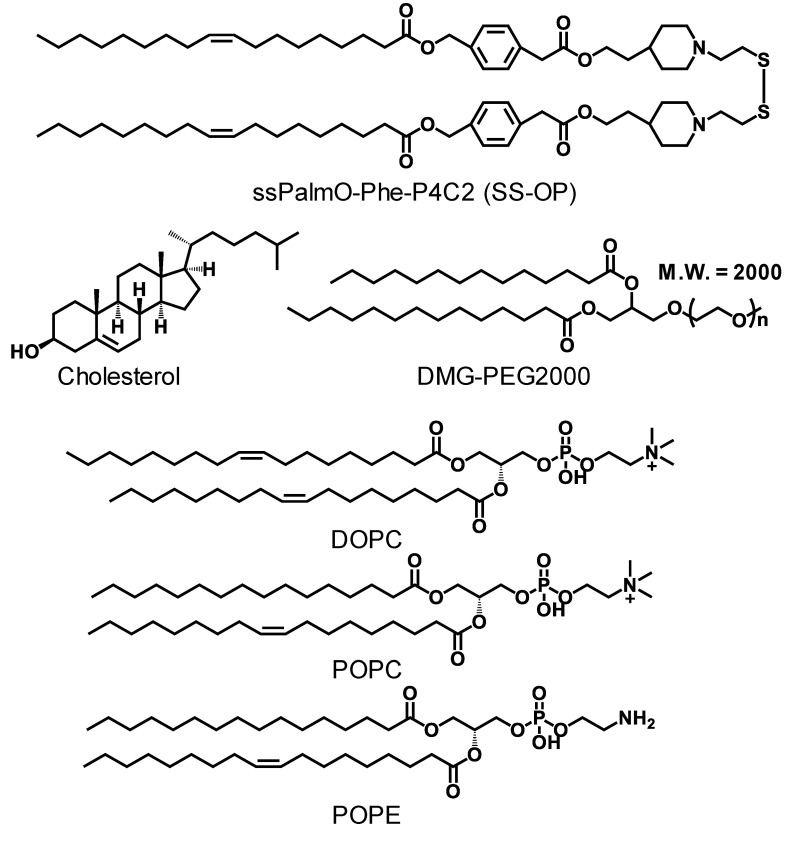
Chemical structures of the lipids used in this study. The chemical structure of the ssPalmO-Phe-P4C2, cholesterol, 1-(Monomethoxy polyethylene glycol 2000)2,3-dimyristoylglycerol (DMG-PEG2000), 1,2-dioleoyl-sn-glycero-3-phosphatidylcholine (DOPC), 1-palmitoyl-2-oleoyl-sn-glycero-3-phosphatidylcholine (POPC), and 1-palmitoyl-2-oleoyl-sn-glycero-3-phosphatidylethanolamine (POPE) were shown.

**Figure 2 pharmaceutics-13-02097-f002:**
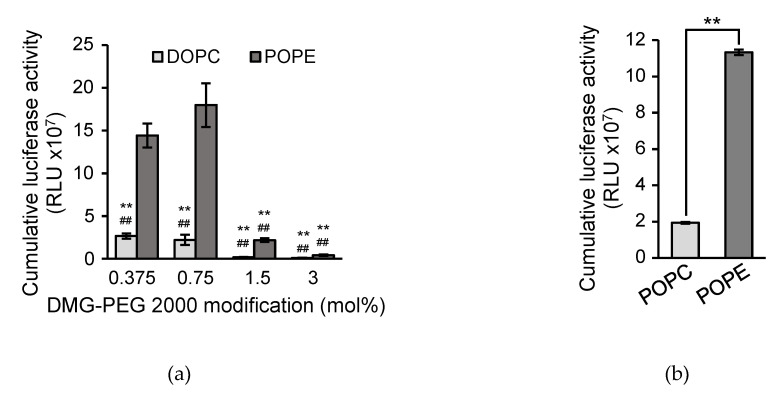
Transfection activity of the LNP_ssPalm_. (**a**) Effects of phospholipids and PEG-lipids on the mRNA transfection activity. the LNP_ssPalm_ with the composition of ssPalmO-Phe-P4C2/DOPC/cholesterol = 52.5/7.5/40 or ssPalmO-Phe-P4C2/POPE/cholesterol = 50/15/30 were prepared with additional 0.375% to 3% of DMG-PEG2000 (*n* = 3). Statistical analyses were performed by the Kruskal–Wallis test followed by Scheffe test, **; *p* < 0.01 against POPE with 0.375% PEG, ##; *p* < 0.01 against POPE with 0.75% PEG. (**b**) Effects of the head group of phospholipids on the mRNA transfection activity. The phospholipid content of the LNP_ssPalm_ was replaced by the POPC to compare the effects of the head group of phospholipids (*n* = 3). Statistical analysis was performed by the student-t’s test, **; *p* < 0.01.

**Figure 3 pharmaceutics-13-02097-f003:**
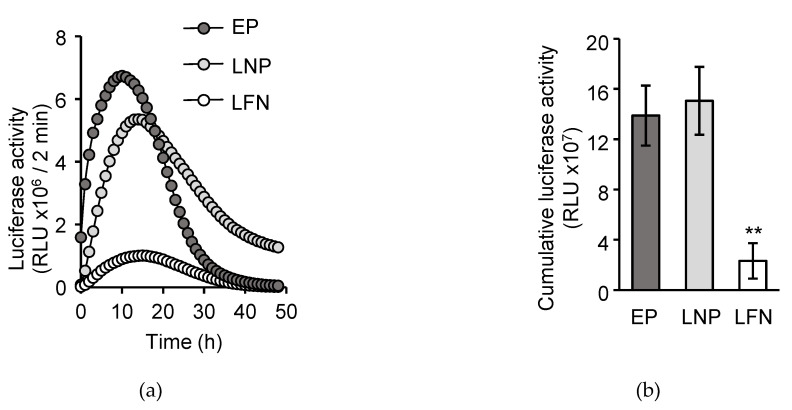
Transfection efficiency of the LNP_ssPalm_ and other transfection methods. (**a**) Comparison with the general transfection procedures. The transfection activity of the LNP_ssPalm_ was compared to the electroporation (EP) or a commercially available lipofection reagent (lipofectamine messengerMAX; LFN). The luminescence from the cultured cells was measured for 2 min in the incubator-type luminometer. Cumulative light unit for 2 min was plotted against time, transfected with mRNA. The mean of the independent experiments is shown (*n* = 3). (**b**) Cumulative luciferase activity. The transfection activity was accumulated for 48 h. Data were represented as mean ± SD (*n* = 3). Statistical analysis was performed by One-way ANOVA followed by Tukey–Kramer test, **; *p* < 0.01 against LFN.

**Figure 4 pharmaceutics-13-02097-f004:**
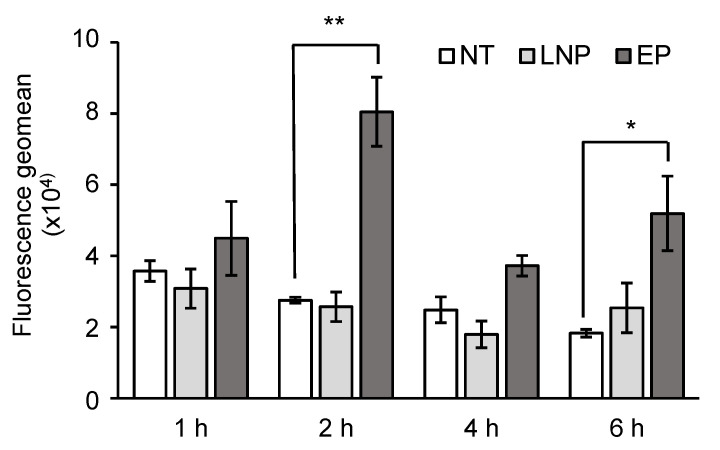
Integrated stress responses induced by the transfection. The phosphorylation level of the eIF2α was compared. Jurkat cells were fixed and permeabilized after the transfection. The intracellular amount of phosphorylated eIF2α was evaluated by an anti-phosphorylated eIF2α antibody. The phosphorylation level of the eIF2α of LNP_ssPalm_ and electroporation (EP) was compared to the non-treated cells (NT). Data were represented as mean ±SD (*n* = 3). Statistical analyses were performed by One-way ANOVA followed by Tukey–Kramer test, *; *p* < 0.05 against non-treated group **; *p* < 0.01 against non-treated group.

**Figure 5 pharmaceutics-13-02097-f005:**
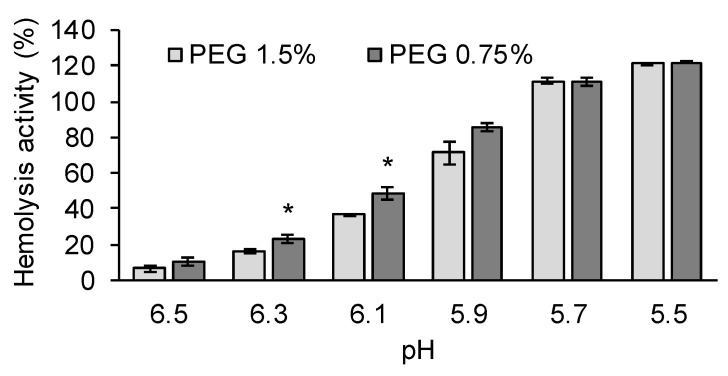
Hemolytic activity of the LNP_ssPalm_ with different amount of PEG-lipids. The LNP_ssPalm_ with 0.75% of DMG-PEG2000 or 1.5% of DMG-PEG2000 was incubated with red blood cells at different pH from pH5.5 to pH 6.5. Data were represented as mean ±SD (*n* = 3). Statistical analysis was performed by student-t’s test, *; *p* < 0.05.

**Figure 6 pharmaceutics-13-02097-f006:**
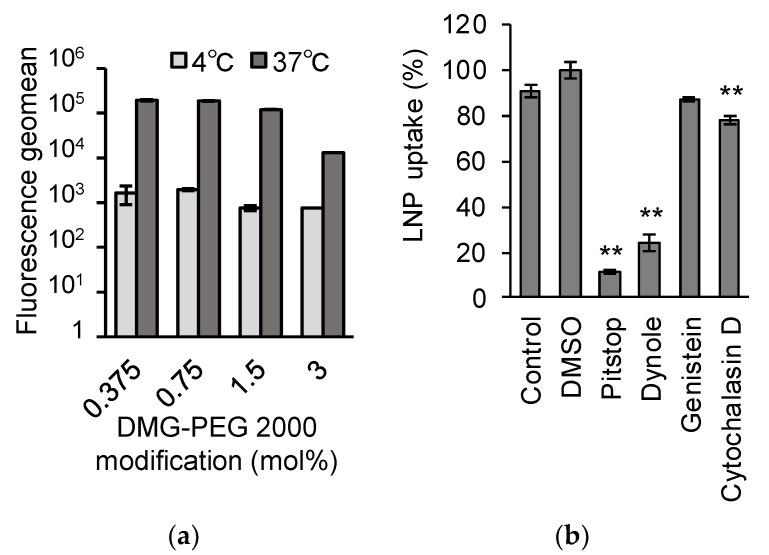
Uptake mechanism of the LNP_ssPalm_. (**a**) The effects of temperature. The Jurkat cells were incubated with the LNP_ssPalm_ at 37 °C or 4 °C. The cellular uptake was evaluated by flow-cytometry. Data are represented as the mean ± SD (*n* = 3). (**b**) pharmacological inhibition of the endocytosis. The Jurkat cells were incubated with LNP_ssPalm_ in the presence of pharmacological inhibitors of endocytosis pathways. As inhibitors, 30 µM Pitstop^®^ 2 (clathrin inhibitor), 20 µM Dynole^®^ 34-2 (dynamin inhibitor), 100 µM Genistein (tyrosine kinase inhibitor), and 10 µM Cytochalasin D (actin inhibitor) were used. Data were represented as mean ±SD (*n* = 3). Statistical analyses were performed by Scheffe’s F multicomparison test (*n* = 3), **; *p* < 0.01.

**Table 1 pharmaceutics-13-02097-t001:** Particle properties of the LNPs in Figure 2.

Helper	PEG Content	Size (nm) ^1^	PdI ^1^	Zeta Potential (mV)^1^	mRNA Encapsulation (%) ^2^
DOPC	0.375%	110.4 ± 8.7	0.15 ± 0.02	−7.16 ± 0.59	73.9 ± 1.6
	0.75%	66.5 ± 5.4	0.10 ± 0.01	−4.47 ± 0.21	84.0 ± 1.0
	1.5%	46.1 ± 4.7	0.08 ± 0.03	−3.15 ± 0.21	85.4 ± 1.1
	3%	36.1 ± 5.1	0.12 ± 0.06	−2.09 ± 0.44	84.7 ± 1.2
POPE	0.375%	106.0 ± 3.6	0.15 ± 0.00	−8.3 ± 0.67	79.0 ± 0.7
	0.75%	63.2 ± 4.3	0.12 ± 0.01	−5.9 ± 0.49	85.8 ± 2.1
	1.5%	45.5 ± 0.6	0.08 ± 0.00	−4.7 ± 0.53	85.6 ± 2.8
	3%	38.9 ± 2.8	0.11 ± 0.01	−2.7 ± 0.57	83.9 ± 1.3

^1^ Size, PdI, and Zeta potential were evaluated by dynamic light scattering. ^2^ mRNA encapsulation was evaluated by Ribogreen^TM^ assay.

## Data Availability

The data presented in this study are available on request from the corresponding author.

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
