# Peer review of "Improvement of mRNA Delivery Efficiency to a T Cell Line by Modulating PEG-Lipid Content and Phospholipid Components of Lipid Nanoparticles"

_pharmaceutics, 2021, doi:10.3390/pharmaceutics13122097_

Round 1
Reviewer 1 Report
This is an interesting description for the uptake of LNPs made with a disulfide containing aid lipid rather than a cationic ionizable lipid (CIL). The principle for the synthesis and performance of the aid lipid was published previously. The new aspects in this draft are a systematic study on the content of PEG-lipid and the exchange between PE and PC phospholipids which seems to improve the uptake in a T cell line where previously no significant uptake was observed. The work deserves publication since it is of interest to the community but there are several points that need to be adressed prior to publication as follows:
The authors argue on the introduction that the surface of the LNP is important for cellular uptake with no reference given. For LNPs made with CILs, a shell-core spherical structure was shown with an asymmetric distribution of the lipidic components.. is it reasonable to assume that for this new formulation based on a different concept the same structure holds? If so, which clues have the authors to assume this?
The study shows that the exchange of PC by PE and the optimisation of PEG-lipids improve the uptake in T cells, but what about in other cell lines? Is the improvement similar to all lines then the argument about selective delivery to T cells falls apart.
I read the cell culture contained serum but the assays for mRNA transfection were not made in the presence of serum. Protein corona in LNPs have been shown to be key for cellular uptake via the ApoE and LDL receptors. In a very recent paper it was shown that ApoE binding induces a change in the distribution of the lipidic components, on the LNP surface... which redistribution of both cholesterol and CIL across the shell and the core. It seems very important to understand whether the results are affected by the presence of serum, as it is expected they will. Therefore some controls should be done in the presence of serum.
Figure 6a shows that the temperature has an effect in the upateke of the particles. It is clear that clathrim mediated endocytosis is relevant. However, lowering the temperature to 4C also affects the membrane properties of the cells making them less fluid. POPE has a melting temperature around 37 degrees. This means that this lipid is fluid at body temperature and in gel phase at 4 C. This will affect the surface properties of LNPs. Controls with DPPE should be taken.
Minor corrections:
the source of cholesterol is not included in the method description
Author Response
Thank you for your reading of our manuscript. Our answers to your comments were summarized in the attached file.

Reviewer 2 Report
Summary: This manuscript reports on an optimized lipid nanoparticle (LNP) formulation, based on a proprietary ionizable lipids, that yielded a 221-fold increase in reporter protein expression in vitro with the Jurkat cell line relative to an unoptimized formulation. The authors systematically replaced the non-non-ionizbale lipids with different molar concentrations of DMG-PEG 2000 and determined POPE with 0.75% DMG-PEG 2000 to be the optimal combination in conjunction with the ionizable lipid, ssPalmO-Phe-P4C2, and cholesterol. This optimized formulation was suggested to better facilitate early endosomal escape and yield transfection levels comparable to electroporation without the induction of an integrated stress response as measured by phosphorylated eIF2α levels. Overall, the manuscript is thorough in the assessment and characterization of the optimized LNPs and contributes towards the progress in transfecting and modifying T-cells. There are, however, a number of comments and suggestions that could be used to improve the manuscript as elaborated below:
MAJOR:
1) While optimizing LNPs in a model T-cell line, specifically Jurkat cells, seems useful, it would be beneficial to comment on the translation of this technology or comparability of Jurkat cells to true T-cells. A connecting experiment, for example, transfecting primary T-cells with the optimized LNP formulation would strengthen the argument for the translatability of the LNPs towards therapeutic development.
2) There is other work recently published to transfect T cells with mRNA LNPs. This is an important area of research though, and this previous work does not take away the impact that current manuscript will have. Examples include:
a) Ionizable Lipid Nanoparticle-Mediated mRNA Delivery for Human CAR T Cell Engineering Margaret M Billingsley et al. Nano Lett. 2020.
b) A commercial product: https://www.precisionnanosystems.com/platform-technologies/genvoy-platform/t-cell-kit-for-mrna
3) As described in the methods, the LNPs used to transfect the Jurkat cells contained mRNA while those used to measure uptake were formulated without mRNA. It would be helpful to address whether the lack of mRNA in the formulations could impact the surface properties of the LNPs such that it would be more difficult to compare the uptake of the particles with the transfection efficiency.
4) While a positive control was described as performed for the hemolysis assay, no controls were shown within the same graphs. A negative control of untreated (no LNP-treated) red blood cells would help to further clarify whether the lysis observed is more of an LNP-dependent effect or a pH-dependent effect. It may also be useful to briefly discuss the limitations of the hemolysis assay as a model for endosomal escape in inferring each LNP’s ability for inducing endosomal release.
5) The conclusion states that: The particles escaped into the cytoplasm at an early stage of endcytosis. However, I do not understand how this can be conclusively stated. I understand that the particles can cause hemolysis, and that this provided potential insights into endosomal escape, but I do not see how conclusive statements about endosomal escape can be made.
MINOR
6) In Supplemental Figure 2, many of the cell viabilities appear to be over 100%. While it does not detract from the overall conclusion regarding cellular toxicity, it may be helpful to address, especially if it is a direct result of the LNPs at low levels acting to help the cells grow or other components in the cell media, such as phenol red, influencing the results of the viability assay.
7) In Figure 1, the chirality of some of the lipids, such as cholesterol and the described helper lipids, appear to be incorrect (wedges instead of dashes).
8) Some very minor suggestions are:
9) There likely needs to be a space between number and the unit for degrees Celsius, such as 37 °C. The symbol for degrees, °, should also be used.
10) The RCF, or G-force, is written as a lowercase g, and typically as × g (e.g.: 500 × g)
11) The concentrations reported are inconsistent and could be standardized to similar units (e.g.: if reporting as /µL or /mL, ensure all units are consistent and not reported as /5 mL).
12) In Figures 2-4, it could be better clarified if the scientific notation, x 104, 106, or 107, is included in the axis title instead of hovering above the graph.
13) In Figure 4, the abbreviations, NT, LNP, and EP, could be defined in the Figure caption.
14) If possible, the units for the y-axis in Figure 3a can also be further clarified or explained in the Figure caption.
15) The kinases, GCN2 and PERK, are first referenced only in the discussion without much preface; it could be useful to incorporate more details, or maybe a diagram of the interactions in the supplementary information, on how they influence eIf4α to help contextualize the rationale.
16) In Figure S21, it would be nice to also see the viability of EP (electroporation) as control.
Author Response

(The authors gave the same response as above.)

Reviewer 3 Report
In this study, the authors report on the development of LNPs with transfection efficiencies that are comparable to that for electroporation in a T cell line (Jurkat cells). It is a very important topic that is well addressed and well written. I recommend the publication of the manuscript after performing the following modifications:
1- Please summarize in the discussion why the incorporation of POPE and DMG-PEG2000 at a concentration below 0.75% showed a high transfection efficiency for the Jurkat cells.
2- Please correct the title of section 3.2.
3- The increase in transfection due to the uptake of the nanoparticles using the clathrin/dynamin-dependent endocytosis was previoulsy proven in:
Informatics in Medicine Unlocked 21 (2020) 100446. Please mention .
Author Response

(The authors gave the same response as above.)

Reviewer 4 Report
- The purpose of the study does not correspond to the title.
- The method of PEGylation is not described well.
- It is not clear how it is prepared these LNPs with different PEG-lipids and phospholipids?
Author Response

(The authors gave the same response as above.)

Round 2
Reviewer 2 Report
The revisions improved the manuscript.
I just have two minor comments.
1) The limitations of the study that Jukart cells were used, rather than primary T cells, should be more clearly stated.
2) The new schematic in the SI did not display correctly when I downloaded the document. It had a lot of non-english text.
Author Response
Thank you for reading our revised manuscript. Our replies are summarized in the attached pdf file.
